# A Therapeutic Vaccine Targeting Rat BORIS (CTCFL) for the Treatment of Rat Breast Cancer Tumors

**DOI:** 10.3390/ijms24065976

**Published:** 2023-03-22

**Authors:** Dmitri Loukinov, Amanda Laust Anderson, Mikayel Mkrtichyan, Anahit Ghochikyan, Samuel Rivero-Hinojosa, Jo Tucker, Victor Lobanenkov, Michael G. Agadjanyan, Edward L. Nelson

**Affiliations:** 1Molecular Pathology Section, Laboratory of Immunogenetics, National Institute of Allergy and Infectious Diseases (NIAID), National Institutes of Health (NIH), Bethesda, MD 20892, USA; 2Center for Immunology, Chao Family Comprehensive Cancer Center, University of California, Irvine, CA 92868, USA; 3Institute for Molecular Medicine, Huntington Beach, CA 92647, USA; 4Natera, Rockville, MD 20850, USA

**Keywords:** brother of the regulator of the imprinted site (BORIS), CT-antigen, breast cancer, vaccine, cancer stem cells, therapeutic efficacy

## Abstract

Cancer testis antigens are ideal for tumor immunotherapy due to their testis-restricted expression. We previously showed that an immunotherapeutic vaccine targeting the germ cell-specific transcription factor BORIS (CTCFL) was highly effective in treating aggressive breast cancer in the 4T1 mouse model. Here, we further tested the therapeutic efficacy of BORIS in a rat 13762 breast cancer model. We generated a recombinant VEE-VRP (Venezuelan Equine Encephalitis-derived replicon particle) vector-expressing modified rat BORIS lacking a DNA-binding domain (VRP-mBORIS). Rats were inoculated with the 13762 cells, immunized with VRP-mBORIS 48 h later, and then, subsequently, boosted at 10-day intervals. The Kaplan–Meier method was used for survival analysis. Cured rats were re-challenged with the same 13762 cells. We demonstrated that BORIS was expressed in a small population of the 13762 cells, called cancer stem cells. Treatment of rats with VRP-BORIS suppressed tumor growth leading to its complete disappearance in up to 50% of the rats and significantly improved their survival. This improvement was associated with the induction of BORIS-specific cellular immune responses measured by T-helper cell proliferation and INFγ secretion. The re-challenging of cured rats with the same 13762 cells indicated that the immune response prevented tumor growth. Thus, a therapeutic vaccine against rat BORIS showed high efficacy in treating the rat 13762 carcinoma. These data suggest that targeting BORIS can lead to the elimination of mammary tumors and cure animals even though BORIS expression is detected only in cancer stem cells.

## 1. Introduction

Cancer immunotherapy has held promise for controlling cancer growth and development for some time. However, this promise has, until recently, remained unfulfilled. With the advent of Immune Checkpoint Inhibitor (ICI) therapy (e.g., monoclonal antibodies against PD-1, CTLA-4, and PD-L1, among others), enthusiasm for anti-cancer immunotherapy has been restored. Unfortunately, ICI therapy elicits clinical benefits (delayed tumor growth and/or improved quality of life) for only a minority of cancer patients [1]. Some cancer patients treated with ICIs experience significant and, in some instances, deadly immune-related adverse effects. The deadliest is immune-related myocarditis, with a mortality rate of about 50% [2]. ICI efficacy has been associated with tumor mutational burden (higher is better), tumor microenvironment hematopoietic cellular profile, and even bowel microbiome. Additionally, various resistance mechanisms to ICIs have been documented to evolve over time [3,4,5]. Thus, there is a rationale for “educating” the immune system in conjunction with ICI treatment in order to maximize the benefit of anti-tumor immunotherapeutic strategies.

The immune system has both a central and a peripheral tolerance for normal cellular elements, proteins, glycoproteins, lipids, etc. Tumor-associated antigens (TAAs) are those antigenic molecules or epitopes that are expressed or over-expressed in tumor cells. TAAs have been characterized as being a ‘slightly mutated self’ [6] and postulated to be relatively poor antigens for stimulating the immune response. This has led various groups to test the efficacy of vaccines targeting whole tumor cells, different tumor-associated antigens (TAAs) [7,8,9] including cancer-testis (CT), antigens [10] and, given the association of tumor mutational burden and ICI efficacy, more recently, neoantigens [11,12,13], in animal models and clinical trials. These various anti-tumor vaccines have had mixed and generally modest success in terms of clinical response rates [13,14]. It should be mentioned that tumor-associated self-tolerance can limit the usage of many TAAs and some CT-antigens [15]. Notably, histologically different tumors can evade the immune responses generated by vaccines targeting TAAs and CT-antigens [16]. One of the exceptions is the Brother of the Regulator of the Imprinted Sites (BORIS, also known as CTCFL), a germline-specific paralog of CTCF—the architectural protein responsible for the three-dimensional structure of the chromatin. Upon abnormal expression in cancer cells, BORIS outcompetes CTCF from its binding sites introducing havoc into the gene expression program, mitosis, and even genome stability [17,18,19]. It was reported that BORIS is localized in the cancer stem cells (CSC) of several human tumor cell lines [20,21]. This, in addition to the observation that BORIS is abnormally expressed in a wide variety of human tumors [18], makes it an attractive molecule to pursue as a target for immunotherapy.

Previously, we reported the partial efficacy of DNA plasmid and Ad5 vector-based vaccines targeting Zinc Finger- (ZF) deleted BORIS (mBORIS) in a 4T1 mouse mammary carcinoma [22,23]. More recently, we demonstrated that DC loaded with mBORIS (DC/mBORIS) induced strong anti-cancer immunity, inhibited 4T1 tumor growth in almost 20% of mice, and dramatically lowered the number of spontaneous congenic metastases [24]. Based on these observations and the important role of BORIS in oncogenic transformation [25], we developed a virus-like replicon particle (VRP), with tropism for dendritic cells [26], and expression of the rat mBORIS target antigen. We demonstrated both the immunogenicity and efficacy of the VRP-based vaccine targeting tumor 13762 in a rat model of mammary adenocarcinoma. Thus, this is a short report showing the effectiveness of monotherapy with the mBORIS therapeutic vaccine for breast cancer treatment. Importantly, this report also demonstrates that targeting a single molecule essential for cancer stem cell survival is sufficient to completely cure the disease in most animals and even prevent disease progression after re-challenge with the same tumor.

## 2. Results

### 2.1. BORIS Expression in the 13762 MAT B III Carcinoma Cell Line

We previously demonstrated that Neu immunotherapy in the rat 13762 MAT B III adenocarcinoma model using VEE-VRP cured 36% of rats with this aggressive breast tumor [27]. However, we also observed down-regulation of rat neu expression and tumor escape with the resumption of rapid tumor growth. Therefore, we hypothesized that targeting the BORIS CT-antigen, an essential factor in maintaining the oncogenic phenotype of cells, with VRP could be an effective treatment while also preventing tumor escape. To use the BORIS antigen in this rat model, we first demonstrated the expression of the *BORIS* gene in the 13762 MAT B III cell line.

Detection of the BORIS expression in the 13762 MAT B III cell line by flow cytometry was challenging due to the absence of a well-validated antibody against BORIS. The first attempts to detect BORIS mRNA by RT-PCR were also unsuccessful until it was shown that BORIS was expressed only in a small side population of tumor cells enriched with cancer stem cells (CSC) [20,21]. Using the same methodology described by Alberti et al. [20,21], we were able to sort around 3 × 10^5^ side population cells from the 13762 MAT B III cell line grown in tissue culture (Figure 1, panels A–C). Using this method, we sorted the so-called “side population” cells intercalating the drug Hoechst 33,345. This drug was pumped away from the cancer stem cell with a verapamil-sensitive multidrug resistance pump. We collected cells that were stained below regular 2n DNA and, as a negative control, contained regular 2n DNA. We detected Boris mRNA in this population by qPCR using two different sets of primers (Figure 1, panels D and E). The expression level was low, suggesting that only a fraction of side population cells express BORIS. mRNA isolated from the testis and the non-sorted 13762 MAT B III cells were then used as positive and negative controls, respectively. Since most housekeeping genes are expressed in the liver at known levels and can be used for quantifying the expression of other genes in any tissue, mRNA isolated from the liver was used for relative quantification.

### 2.2. Generation of a VRP-BORIS Vaccine

We previously developed an immunogen comprising a BORIS molecule that lacked the DNA-binding Zinc Fingers (ZF) domain (mBORIS) in order to avoid safety concerns associated with functional BORIS. First, it was shown that aberrantly expressed functional BORIS could transform NIH3T3 cells [25]. Second, the BORIS ZF domain showed high homology and similarity with other ZF-proteins that are the most abundant family of the expressed genes [28]. Thus, BORIS can induce off-target immune responses specific to various proteins.

The schematic cloning strategy of the rat BORIS vaccine is shown in Figure 2 panel A. Briefly, PCR fragments of N- and C-termini of rat BORIS were fused through a flexible linker that allows independent folding of those parts upon expression. Modified *BORIS* (*mBORIS*) was cloned into the pET24d *E. coli* expression vector for recombinant protein purification (Figure 2 panel B) and into a VEE-based vector to generate virus-like replicon particles (VRPs). The VRP is a propagation-deficient, single-cycle viral vector in which some of the viral sequences encoding the structural proteins have been replaced with sequences encoding the rat mBORIS. The VEE-derived VRP is also the only vector with selective and restricted tropism for a subset of DCs both in vitro and in vivo [26,29,30]. The VRP has shown great promise in studies targeting infectious diseases (reviewed in [31]), and there have also been reports of anti-tumor immunotherapy applications targeting various tumor-associated antigens (TAAs) [32,33,34,35,36,37,38,39,40]. These positive reports, and the ability of this vector system to elicit CTL/Th1 biased immune responses [31,41,42], support the potency and efficacy of the VRP vector system.

The expression of VRP-encoded mBORIS was shown in BHK cells transduced with VRP-mBORIS or VRP-EGFP as an irrelevant control. As shown in Figure 3 panel A, VRP structural protein nsP2 was detected in cells transduced with either VRP-mBORIS or VRP-EGFP, while mBORIS was detected only in cells transduced with VRP-mBORIS. None of these proteins were detected in non-transduced BHK cells. GAPDH served as a loading control. mBORIS mRNA expression was demonstrated by real-time qPCR. BORIS expression was only detected in cells transduced with VRP-mBORIS. No mRNA specific to mBORIS was detected in non-transduced cells (Figure 3 panel B).

### 2.3. Evaluation of Therapeutic Potency of VRP-mBORIS Vaccine

We tested the therapeutic efficacy of VRP-based immunotherapy targeting the CT-antigen BORIS in rats who had received the heterotopic inocula of the 13762 MAT B III tumor cells 48 h earlier. Rats were injected with VRP-mBORIS, or formulation buffer (no treatment), as shown in Figure 4 panel A. Tumors became palpable by approximately day 7 in all animals. Control animals, those that had received formulation buffer alone, all rapidly succumbed to the large tumor volume requiring animal euthanization by day 25 (Figure 4 panel B). In contrast, in 50% of the animals that had received VRP-BORIS therapy, we observed complete tumor regression by approximately day 42. Survival (Kaplan–Meier) curves representative of 3 experiments are shown in Figure 4 panel C. Animals were followed for at least 100 days after tumor inoculation, and those that demonstrated complete tumor regression did not relapse throughout this time. These data suggest that VRP-based anti-tumor immunotherapy targeting the CT-antigen BORIS inhibits tumor growth and prolongs the survival of a rat mammary tumor model.

Moreover, evidence for immunological memory could be ascertained by performing tumor re-challenge experiments in VRP-treated animals that had survived their original tumor burden. To this end, the surviving animals were challenged with the same dose of the 13762 cells in the right rear flank (the opposite side from the original tumor site) more than 144 days after the original tumor inoculation (Figure 4 panel D). These animals were monitored for tumor growth as previously described. All of the animals that had rejected their initial tumor did not develop palpable tumors or succumb to the second tumor challenge. However, all naive animals developed tumors requiring their euthanization as expected. These data provide evidence for immunological memory in animals receiving VRP-mBORIS.

To evaluate the cellular immune responses in rats cured of the tumor, we measured the splenic T cell proliferation rate and the number of splenocytes producing IFNγ in response to in vitro restimulation with recombinant rat BORIS. As shown in Figure 4 panel E, cells from the VRP-mBORIS immunized and cured rats showed a significantly higher proliferation rate in response to the recombinant mBORIS protein than splenocytes from the naïve rats (*p* < 0.05). The number of cells producing IFNγ was significantly higher in the splenocytes from immunized rats restimulated in vitro with the recombinant mBORIS protein compared to splenocytes restimulated with irrelevant GFP derived from *E. coli* or splenocytes from the non-immunized control rats (Figure 4, panel F).

## 3. Discussion

Recent advances in understanding the molecular biology and immunology of cancer provide new insights for the generation of new anti-cancer vaccines and therapeutics, targeting antigens that are not expressed in somatic cells but become abnormally expressed in cancer cells (e.g., CT-antigens). The immune system recognizes CT-antigens as foreign molecules upon immunization [43,44,45]. Previously, using the mouse 4T1 breast cancer model, we have shown that induction of strong CTL response against CT-antigen BORIS leads to the eradication of tumors and a dramatic decrease in metastatic disease in vivo [23,24].

In this paper, we report using rat BORIS as a target for the immunotherapy of the rat mammary carcinoma 13762. We used the recombinant construct of rat BORIS with 11 Zinc Finger DNA-binding domains of BORIS substituted with the flexible linker, allowing independent folding of N- and C-terminal domains in the alphavirus vector as a therapeutic vaccine. We showed that all rats in the control and experimental groups developed tumors, but in the group receiving the BORIS-containing vaccine, up to 50% of animals became cured. In groups receiving no treatment, all rats succumbed to the tumor. Animals that survived were watched for three months, and no tumor recurrence was observed. Moreover, even the re-challenge of cured animals with a new injection of the 13762 tumors showed no palpable tumor development and had 100% survival during the observation time, suggesting the long-lasting memory and potency of the immunotherapeutic intervention. No adverse effects of treatment were noted.

As we found that BORIS was expressed only in a fraction of the “side population” cells considered CSCs, this study indicates that eliminating a small fraction of those cells can be sufficient to cure rats of the 13762 mammary carcinoma. This observation is in concordance with the “cancer stem cell” theory, which suggests that the CSC phenotype is necessary to maintain the aggressive nature of the tumor, including the spreading of distant metastasis, and, without them, the tumor loses its aggressive nature and gradually disappears [46]. To date, there is no known universal marker for CSCs. We speculate that BORIS might be such a marker. However, additional research is needed to confirm this data along with that of others in support of this hypothesis [18,20,21,47,48].

Various tumor-associated antigens (TAA) have been used as targets to drive anti-tumor immune response for a relatively long time with limited success [49]. Several obstacles limit the therapeutic success of anti-TAA vaccines. First is the self-tolerance of the host immune system to TAAs [49]. The second is immune evasion. Although tolerance can be overcome by various means of inducing an immune response, TAAs can be modified by the tumor to alter epitopes and evade attack by the host’s immune system. Tumor mutation rates may correlate with response to immunotherapy [50]. TAA expression may even be completely shut down for the same purpose of evasion of immune targeting. Third, the immune response to the target antigen may be potentially harmful to the patient if the antigen cross-reacts with epitopes of non-malignant tissues, thus causing dangerous auto-immune reactions. These factors make the choice of TAAs crucial for the success of the therapeutic anti-cancer vaccine.

Theoretically, the ideal target should be: (1) restricted to the tumor and not to be expressed outside of the tumor except in immune-privileged sites, (2) be able to induce a strong CTL response upon vaccination, and (3) be absolutely essential for tumor growth, such that neoplastic cells must depend on it for successful expansion, and its loss should either kill the tumor cell or revert its neoplastic phenotype. BORIS is such an antigen. BORIS was identified as a mammalian paralogue of CTCF- a human architecture protein responsible for chromatin 3D organization [51]. It appeared in evolution by duplication of CTCF probably during tetraploidization in fish and became germ cell-specific in mammals. This transcription factor, aberrantly expressed in cancer, competes for binding sites with its somatic analog CTCF, alters the epigenetic landscape, and even induces the transformation of mouse fibroblasts during ectopic expression [18,25]. BORIS can also regulate the expression of other CT-antigens, including those used in clinical trials as targets for tumor vaccines such as MAGE-A1, NY-ESO-1, etc. [52,53,54,55,56]. In two rodent breast cancer models, mouse 4T1 and rat 13762, a significant proportion of animals were cured by the therapeutic anti-BORIS vaccine. Not all animals were cured, and we speculate that in these cases, the balance between tumor-associated immune suppression and anti-tumor immune activation tipped toward immune suppression. The potential factors that lead to this imbalance are numerous, and defining them is beyond the scope of this report [57]. However, the absence of tumor recurrence in the cured rats in this current study supports the ability to maintain an anti-tumor immune response if the therapy can overcome the factors that amplify tumor immune suppression.

With the success of liposome-delivered mRNA vaccines for SARS-CoV-2, enthusiasm for RNA vaccines has reached an all-time peak [58]. The capacity to elicit both humoral and cellular immune responses against this viral pathogen highlights the advantages of this approach. However, pathogen antigens are not the same as tumor-associated antigens. Traditional vaccination approaches have not successfully elicited effective anti-tumor immune responses. In contrast, the VEE-VRP platform has a tropism for immune system dendritic cells [26,29,59,60], the most potent antigen-presenting and immune stimulating cells of the immune system, suggesting that the VEE-VRP is a particularly attractive vaccine delivery platform.

Our data support the capacity of VRP-BORIS to educate the immune system for an effective anti-tumor immune response. The capacity to eliminate previously established tumors in 50% of the animals suggests that additional adjunctive maneuvers could increase this efficacy/activity, including chemotherapy, radiation therapy, molecularly targeted therapies, and immune checkpoint inhibition. Which, if any, of these adjunctive maneuvers will prove to be most advantageous is a question for future studies. Regardless, the potency of the VRP platform indicates that this is a solid foundation for further therapeutic development.

The efficacy of the alphavirus-based therapeutic cancer vaccine against BORIS was comparable, if not better, to VRP HER2neu tested under the same conditions [27]—a target already in clinical use. Overall, the ability of a therapeutic cancer vaccine targeting BORIS to cure a rat model of breast cancer raises hope that a similar vaccine could successfully treat cancer in humans. That is why we are planning to start the Phase 1 clinical trial of a therapeutic vaccine using BORIS as a target to treat triple-negative breast cancer. We expect the vaccine to be safe and hope to register improvement in tumor and metastasis load in immunized subjects.

## 4. Materials and Methods

### 4.1. Animals and Cell Lines

Approximately 6–8 week-old Fisher 344 female rats were purchased from NCI FCDC and housed in grouped cages under normal vivarium conditions. The rat mammary tumor cell line, 13762 MAT B III (CRL-1666, ATCC), was obtained and cultured in vitro as recommended by ATCC. Thereafter, the 13762 MAT B III cells were split 50:50 approximately 24 h before use. Cells were harvested and washed at least twice in phosphate-buffered saline prior to suspension in sterile PBS for inoculation into recipient animals. All experiments were conducted in accordance with a University of California, Irvine, Animal Care and Use Committee (UCI-IACUC) approved protocol and in accordance with all Federal and local regulations.

### 4.2. Side Population Flow Cytometry and Sorting

The 13762 cells grown in culture were collected by centrifugation (1200 g, 10 min), diluted in a small quantity of cell culture media without serum, and stored on ice. Cells were incubated for 60 min at 37 °C, filtered through a cell strainer of 40 μm mesh size, and subjected to either analysis or sorting. Before sorting, 10^6^ cells/mL were stained with Hoechst 33342 at 12.5 μg/mL. Analysis was performed on LSRII (Becton Dickinson, Franklin Lakes, NJ, USA), and sorting was performed on FACS Aria at the NIAID flow cytometry facility. Verapamil at 50 μM concentration was used as a multidrug pump blocker for analytical purposes.

### 4.3. Generation of VRP and Recombinant Protein Reagents

VRPs containing coding sequences for the heterologous proteins were produced by AlphaVax, Inc. (Research Triangle Park, Durham, NC, USA) using established methods based on those previously described [61]. VRPs were resuspended in formulation buffer (10 mM sodium phosphate buffer, pH 7.4, containing 1% normal rat serum and 5% sucrose), shipped on dry ice, and stored at −80 °C until use. VRPs were diluted in formulation buffer to an appropriate concentration for administration. Control animals received injections of formulation buffer alone or with formulation buffer containing VRP encoding and irrelevant antigen.

The rat BORIS gene was amplified from Rat Testis Marathon^®^-Ready cDNA (Takara, San Jose, CA, USA) and cloned into the PCR4-TOPO vector (Invitrogen, Carlsbad, CA, USA). For the generation of ZF-deleted modified BORIS (mBORIS), regions of genes encoding N-terminal and C-terminal domains of *BORIS* were PCR-amplified and linked together through a specifically designed short spacer.

For mBORIS recombinant protein production, the *mBORIS* gene was cloned in frame with C-terminal His-tag into a pET24d vector (Novagen, Madison, WI, USA). For in vivo expression and large-scale purification, *E. coli* BL21 (DE3) was transformed with the pET24d-mBORIS plasmid and grown in Luria broth with kanamycin (28 °C, OD 600 = 0.8). Protein synthesis was induced by isopropyl-D-thiogalactopyranoside (1 mM, 3–5 h; Calbiochem, San Diego, CA, USA). Rat BORIS protein was purified using a Nickel-agarose column (Qiagen, Germantown, MD, USA.). Positive fractions were combined, dialyzed against PBS, and concentrated with Amicon Ultra centrifugal filter devices (50,000 MWCO; Millipore, St. Louis, MO, USA).

### 4.4. Detection of VRP-Encoded mBORIS by Western Blotting

BHK-21 (CCL-10, ATCC) cells were transduced at a multiplicity of infection (MOI) 5–10 for one hour at 37 °C, then transferred to tissue culture flasks and incubated for 16–18 h at 37 °C, 10% CO_2_. Cells were harvested with trypsin. One part of the cells was kept for RNA isolation (see below), and another part of the cells was lysed by incubating the cells for 30 min with a standard RIPA lysis buffer containing the protease inhibitors, aprotinin (5 μg/mL), leupeptin (5 μg/mL), pepstatin A (5 μg/mL), and PMSF (1 mM) as previously described [26]. The cell lysate was centrifuged at 12,000× *g* for 10 min at 4 °C. The resulting cell lysates were stored at −80 °C until use. Cell lysates were mixed with 4X NuPAGE LDS sample buffer, 10X NuPAGE reducing agent, and either PBS or water and heated to 50 °C for 10 min. Samples were then loaded onto NuPAGE 10–12% Bis-Tris gels (Invitrogen) with MOPS running buffer (Invitrogen) and transferred onto Hybond-ECL nitrocellulose membranes (GE Healthcare). Membranes were blocked with 5% milk in TBS-tween and then incubated with the appropriate primary antibody. Membranes were washed and incubated with the appropriate HRP-conjugated secondary antibody (Jackson ImmunoResearch, West Grove, PA, USA). After additional wash steps, membranes were incubated in ECL or ECL Plus detection reagent (GE Healthcare, Chicago, IL, USA) according to the manufacturer’s instructions. The signal was detected after membranes were exposed to HyBlot CL autoradiography film and developed. Membranes were stripped with Restore Western Blot Stripping Buffer (Pierce, Appleton, WI, USA) or 0.2 M NaOH, washed, blocked, and probed for additional proteins in the same manner.

### 4.5. Detection of VRP-Encoded mBORIS mRNA by Real-Time qPCR

VRP-transduced BHK-21 cells were lysed in Trizol (Invitrogen, Carlsbad, CA, 92008-7313 USA). RNA was isolated according to the manufacturer’s protocol, and the RNA concentration was determined using a NanoDrop ND-1000 spectrophotometer (NanoDrop Technologies, Wilmington, DE, USA). The resulting RNA was DNase treated (Stratagene, La Jolla, CA, USA), and equal amounts were used to generate cDNA using Oligo dT (Promega, Madison, WI, USA) or gene-specific primers (Eurofins MWG Operon, Huntsville, AL, USA) and SuperScript III Reverse Transcriptase (Invitrogen Carlsbad, CA, USA) according to the manufacturer’s instructions. A no-RT control was performed for each sample as well. After the cDNA synthesis step, the resulting cDNA was treated with 1 μL of RNase cocktail (Ambion, Austin, TX, USA) and stored at −20 °C until use in qPCR.

Probes and primers were designed using Roche’s free web-based software (https://www.roche- applied-science.com) and are as follows: VRP-rat BORIS forward: 5′-AGA-CCC-CGG-ACT-TTC-CAG-3′, VRP-rat BORIS reverse: 5′-AGG-GGG-CTG-ACT-TCT-CG′3′, rat β-actin forward: 5′-CTC-CCT-CAT-GCC-ATC-CTG-3′, rat β-actin reverse: 5′-GTA-GCC-ACG-CTC-GGT-CAG-3′. Probes were chosen from the Roche Universal Probe Library (UPL) for the respective primer pairs and are as follows: BORIS Probe #22, β-actin: Probe #20. A LightCycler 480 QPCR system (Roche, Indianapolis, IN, USA) and the LightCycler 480 Probes Master Mix kit (Roche, Indianapolis, IN, USA) were used for real-time QPCR assays, according to the manufacturer’s instructions, with the following amplification protocol: 1 pre-incubation cycle at 95 °C for 5 min, followed by 55 cycles of 95 °C for 10 s, 60 °C for 30 s, and 72 °C for 1 s. Data are presented in relative expression units, which were calculated by the formula 2^−ΔCt^, where ΔCt = Ct of the gene of interest—Ct of the housekeeping gene as discussed in [62].

### 4.6. Real-Time PCR

Total RNA was isolated using the RNeasy Minikit (Qiagen) according to the manufacturer’s protocol, and cDNA was prepared using 1 µg of RNA and the SuperScript III first-strand synthesis system (Thermo Scientific, Waltham, MA, USA) according to the manufacturer’s protocol. Real-time PCR was performed using the TaqMan™ Universal PCR Master Mix (Thermo Scientific Waltham, MA, USA) and a 7900HT sequence detection system (Applied Biosystems Waltham, MA, USA) with the following amplification protocol: 1 pre-incubation cycle at 95 °C for 5 min, followed by 40 cycles of 95 °C for 10 s, 60 °C for 30 s, and 72 °C for 30 s. The primers and probes (TAMRA/FAM) were designed using the PrimerQuest Tool (IDT DNA Technologies, San Diego, CA, USA) and are as follows: rat Boris Set 1:

forward: 5′-CAGGGAGGGACAGATAAGAAAG-3′,

reverse: 5′-GAGGTGGGAAATGGTCAGAA-3′,

probe: 5′-TCGTCTTTCCCTTCTCTTGGTTTCATTTCC-3′;

rat Boris Set 2: forward: 5′-GGAGGAGATCAAGTGCAGATAC-3′,

reverse: 5′-CTTCTCATTCTTGTGGGTCCTC-3′,

probe: 5′-CACGAACGCTATGCCCTCCTTCAG-3′;

rat Gapdh: forward: 5′-ACTCCCATTCTTCCACCTTTG-3′,

reverse: 5′-CCCTGTTGCTGTAGCCATATT-3′,

probe: 5′-TTGTCATTGAGAGCAATGCCAGCC-3′.

Data are presented in relative expression units, which were calculated by the formula 2^−ΔCt^, where ΔCt = Ct of the gene of interest—Ct of the housekeeping gene (Gapdh) as discussed in [62]. Rat testis and liver RNA were used as a positive and negative control, respectively.

### 4.7. Tumor Inoculation and VRP Immunotherapy Administration

Before tumor inoculation, tumor cells were phenotyped as described above and tested for mycoplasma according to the manufacturer’s instructions (Lonza, Allendale, NJ, USA). Mycoplasma negative tumor cells (1 × 10^5^ viable cells) were administered into the subcutaneous space on the left flank, approximately 1 cm cephalad, and lateral to the base of the tail. Animals were immunized every 10 days starting ≥48 h after tumor inoculation, with 200 μL of formulation buffer, with or without VRP, and administered through a 27-gauge needle in cohorts of 6–12 animals, depending on the experiment. After the administration site was cleansed with 70% ethanol, all injections were performed using minimal restraint of the conscious animals. Subcutaneous VRP immunizations were located in the hindquarter approximately 1–2 cm, either cephalad or caudal, to the tumor inoculation site. Immunotherapy was discontinued after 4–6 administrations. Tumor volumes were regularly measured and calculated using the formula: volume = 0.4 (ab^2^), where “a” and “b” represent perpendicular axis measurements with “a” representing the longest axis dimension [63]. Tumors exceeding 10,000 mm^3^ or any signs of distress in the animals were indications for euthanization, which was performed by CO_2_ inhalation. For in vitro immunological assays, non-tumor-bearing animals received three VRP immunizations every 10 days before euthanization and tissue harvest in order to isolate immune cells by mechanical disruption of lymphoid tissues.

### 4.8. Proliferation Assay

Spleens were isolated and disrupted mechanically into single-cell suspensions. Red blood cells were lysed with ACK lysis buffer (Lonza, Allendale, NJ, 07401, USA). Cells were washed to remove any free proteins, resuspended at 10^7^ cells/mL in PBS, and incubated with 5 μM CFSE (Invitrogen, Carlsbad, CA, 92008-7313 USA) in the dark for 8 min at room temperature. Cells were washed with complete media three times to eliminate unincorporated CFSE. The cells were then restimulated with 160 μg/mL recombinant rat BORIS protein or a GFP control protein (with the same level of endotoxin as the recombinant rat BORIS) and incubated at 37 °C. After 96 h in culture, cells were harvested, and proliferation was assessed by measuring the dilution of the CFSE signal using flow cytometric means. The dye 7-AAD (BioLegend, San Diego, CA, USA and BD Biosciences, San Jose, CA, USA) was utilized to exclude the dead cells from the subsequent analysis of cells that had undergone at least one round of proliferation.

### 4.9. ELISpot Assay

The same splenocytes isolated for proliferation assay as described above were used for ELISpot assays to determine the number of antigen-specific cells producing IFNγ. Cells were restimulated with 20–160 μg/mL recombinant rat BORIS protein or with GFP as a control protein. As a positive control, cells were stimulated with 2.5 μg/mL pokeweed mitogen (Sigma-Aldrich, Saint Louis, MO, USA). ELISpot assays were performed according to the manufacturer’s instructions (MabTech, Cincinnati, OH, USA). Spots were counted using a CTL-ImmunoSpot S5 Macro Analyzer (Cellular Technology Ltd., Shaker Heights, OH, USA), and the difference in the number of spots per 10^6^ splenocytes in restimulated minus non-re-stimulated samples was calculated.

## 5. Conclusions

In conclusion, we have demonstrated that the VRP-based anti-rat BORIS vaccine has therapeutic efficacy in a treatment setting. This vaccine strategy inhibited the growth of the pre-existing, aggressive, and difficult-to-eradicate 13762 mammary adenocarcinomas. The vaccine elicits strong and potent anti-cancer immunity, helping eliminate pre-existing tumors and prevent tumor development upon re-challenge. No noticeable adverse effects have been observed upon vaccination.

Although BORIS was expressed in the 13762 tumors at a very low level only in cancer stem cells, the immune system’s eradication of BORIS-expressing cells appeared to be sufficient to cure relatively large-volume tumors in murine and rat models. We anticipate that the efficacy of targeting BORIS may be recapitulated in human trials, leading to a safe and successful anti-tumor immune response.

## Figures and Tables

**Figure 1 ijms-24-05976-f001:**
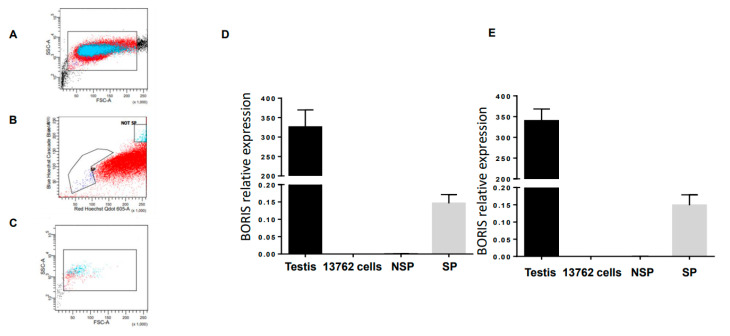
Boris expressed in the side population of rat mammary tumor cell line, the 13762 MAT B II, presenting CSC. (**A**–**C**) Flow cytometry and sorting gates for side population (SP) and non-side population (NSP). (**D**,**E**) Total RNA from the 13762 MAT B II cells, SP (**D**), and NSP (**E**) isolated from the 13762 MAT B II cells, testis, and liver were extracted, and BORIS expression was analyzed by qRT-PCR. The results were normalized to GAPDH expression and related to BORIS expression in liver cells. Error bars represent the mean ± SD (n = 3). Two sets of BORIS primers have been used, set 1 (**D**) and set 2 (**E**).

**Figure 2 ijms-24-05976-f002:**
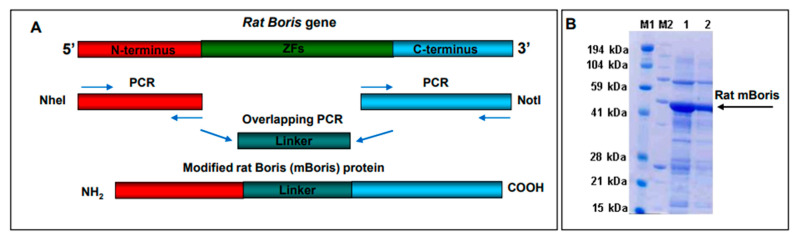
Schematic representation of the cloning strategy of rat ZF-deleted modified BORIS (mBORIS). The N-terminus region and C-terminus encoding regions of the *BORIS* gene were amplified by PCR and linked together through the flexible linker using HindIII and BamHI-introduced restriction sites (**A**). The amplified gene was cloned into the pET-24a *E. coli* expression vector. mBORIS recombinant protein was purified from *E. coli* BL21 (DE3) and analyzed in PAGE (**B**).

**Figure 3 ijms-24-05976-f003:**
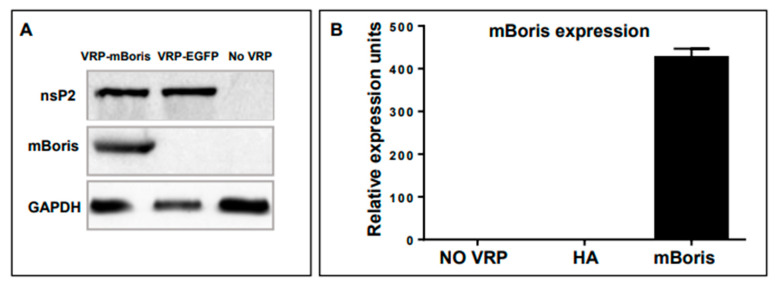
VRP-encoded mBORIS are expressed by transduced BHK cells. (**A**) Transduced cells were lysed with RIPA lysis buffer containing protease inhibitors. Lysates were assessed for VRP-encoded protein expression by Western Blot. All lysates from cells transduced with VRP expressed the VRP non-structural protein nsP2. (**B**) RNA was isolated and used for first-strand cDNA synthesis. The expression of VRP-encoded mBORIS was determined by real-time qPCR. Expression of a particular VRP-encoded tumor antigen sequence was observed only in BHK cells that had been transduced with VRP-encoding that particular sequence. Data are representative of >3 experiments.

**Figure 4 ijms-24-05976-f004:**
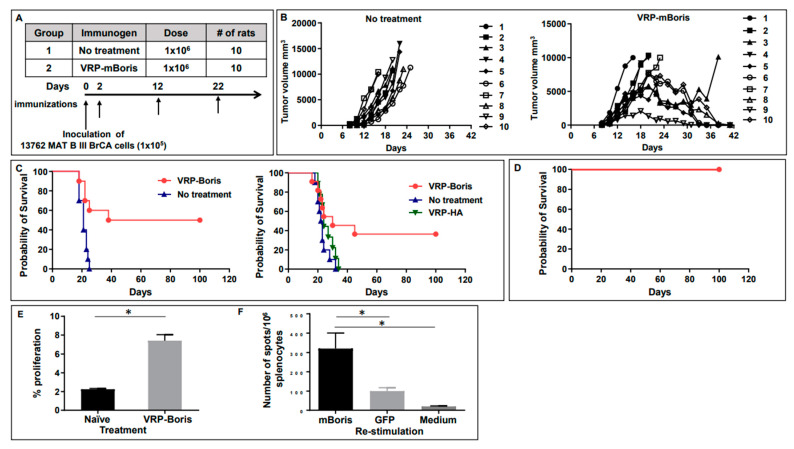
Therapeutic vaccination with VRP-mBORIS inhibits tumor growth and cures rat breast cancer, inducing antigen-specific cellular responses. (**A**) The experimental design of the therapeutic study. Rats were treated with VRP-mBORIS 2 days after injection of the 1 × 10^5^ 13762 MAT B III BrCA cells. (**B**) Depicts the tumor size of individual animals on the designated days for VRP-mBORIS treatment and non-treated animals. (**C**) Depicts survival of rats in two independent experiments. In one experiment, rats were also treated with irrelevant therapeutics VRP-HA. It was observed that 38–50% of rats were completely cured of the disease, while rats without treatment or treated with irrelevant immunogen died on days 25–34. (**D**) Survived mice were re-challenged by inoculation of the 1 × 10^5^ 13762 MAT B III BrCA cells. It was observed that 100% of rats were resistant and did not develop a new tumor. (**E**) Proliferation upon in vitro restimulation with recombinant mBORIS protein. Splenocytes labeled with CFSE were restimulated in vitro with 160 µg/mL recombinant rat Boris or recombinant GFP in the presence of 50 U/mL IL-2. Four days later, cells were collected and analyzed for proliferation by dilution of the CFSE signal using flow cytometry. Data are represented here as the % of cells that had undergone at least one round of proliferation in response to stimulation with recombinant mBORIS protein—the % of cells that had undergone at least one round of proliferation in response to stimulation with the control protein, recombinant GFP. Data depicted here were generated from cohorts of three animals. Error bars represent standard error, * indicates *p* < 0.05. (**F**) IFNγ producing splenocytes were detected after in vitro restimulation with indicated proteins by ELISPOT assay.

## Data Availability

All data generated or analyzed during this study are included in this article, and materials are available from the corresponding author upon reasonable request.

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
