# Peer review of "A Therapeutic Vaccine Targeting Rat BORIS (CTCFL) for the Treatment of Rat Breast Cancer Tumors"

_ijms, 2023, doi:10.3390/ijms24065976_

Round 1
Reviewer 1 Report
The manuscript described a new immunological modality to treat cancer. The results were encouraging.
1. In section 2.1, why did the authors select “mRNA isolated from the liver” for relative quantification concerns? Did the relevant housekeeping genes of the tissue work?
2. Did the isolation of side population cells base on stem cell markers? Had the authors tested the stemness of the isolated cells?
3. To test the hypothesis of immunological memory, the authors selected the rats 144 days after the original tumor inoculation. What was the percentage of survival for the tumor burden vaccinated rats? Were the authors sure that 144 days were enough for testing immunological memory?
4. The first 3 sentences in section 3 are very strange. They seemed to be a reviewer’s comments.
Author Response
First of all, we would like to thank the reviewer for considering our manuscript, ID: ijms-2246057, "A therapeutic vaccine targeting rat BORIS (CTCFL) for treatment of rat breast cancer tumor" for publication in the International Journal of Molecular Sciences in the special issue " Tumor Microenvironment and Immune Response in Breast Cancer." As requested, we are submitting the revised manuscript, and this letter describing our responses to the concerns raised by the reviewer addressed point-by-point below.
First reviewer:
- In section 2.1, why did the authors select "mRNA isolated from the liver" for relative quantification concerns?
Most housekeeping genes are expressed in the liver at known levels. Therefore, they can be used for the relative quantification of mRNA of other genes in any tissue. This information was added to the revised manuscript (page 3, lines 110-112).
- Did the relevant housekeeping genes of the tissue work?
Yes, it did (please see the response above).
- Did the isolation of side population cells base on stem cell markers?
Specific stem cell markers are used in human breast cancer and sometimes in mouse breast cancer. Unfortunately, no stem cell markers are known for rat breast cancer cells. Instead, we used sorting of the so-called "side population" cells intercalating drug Hoechst 33345. This drug is pumped away from the cancer stem cell with verapamil sensitive multidrug resistance pump. We collect cells that are stained below regular 2n DNA and, as a negative control, contain regular 2n DNA This information was added to the revised manuscript (page 3, lines 103-106) .
- Have the authors tested the stemness of the isolated cells?
Thanks, it is an important question. Unfortunately, this is not an easy task, specifically for rat stem cells. Therefore, we relied on published data that the "side population" contains cells that are enriched for cancer stem cells for multiple cancers in different mammalian species (please see references 11 and 12).
- 5. To test the hypothesis of immunological memory, the authors selected the rats 144 days after the original tumor inoculation. What was the percentage of survival for the tumor burden vaccinated rats?
In Figure 4, panels C and D, we showed that up to 50% of rats survived after primary tumor inoculation, but the death likely occurred because the tumor grew faster than the immune response developed. However, after the second tumor challenge survival rate was 100%, and no significant tumor growth was detected upon the second tumor inoculation.
- Were the authors sure that 144 days were enough for testing immunological memory?
144 days are almost 1/6 of the average lifespan of rats. In humans, vaccines effective for 10 years are considered very effective, so the term 144 days is comparable to roughly 10 years of memory in humans.
- The first 3 sentences in section 3 are very strange. They seemed to be a reviewer's comments.
Sorry, it was a typo. Those three sentences were deleted.
We have tried to address first reviewer's concerns fully and revised our manuscript accordingly.
If you have any questions regarding this manuscript, please contact me directly. For your convenience, my cell phone number is 301-830-0110, and my e-mail address is dloukinov@niaid.nih.gov.
Thank you for your attention to our manuscript.
Sincerely,
Dmitri Loukinov, Ph.D.

Reviewer 2 Report
This study demonstrated that the VRP-BORIS vaccine was effective in inhibiting tumor growth and prolonging the survival of rats with mammary tumors. The vaccine-treated rats showed evidence of immunological memory and a stronger cellular immune response compared to control rats.
However, authors used a limited number of rats for each treatment group, which may not fully reflect the variability and heterogeneity of the disease. The study also only provided information about the effects on a single type of breast cancer and its effects on other types of cancer are not known. Further, the study dismissed the comparison between VRP-BORIS vaccine to other treatment options (e.g. chemotherapy or radiation therapy models, or any applicable), which would provide a more comprehensive evaluation of its therapeutic efficacy in comparison to established treatments. Moreover, the results only show the short-term effects of the vaccine, and it is not clear if the benefits will be sustained over the long-term. Likewise, characterizing the specific immune response elicited by the vaccine, including which specific immune cells are activated and the types of cytokines produced, would provide additional insight into how the vaccine works and how it could be optimized. Thus, the mechanism by which the vaccine induces an immune response and eliminates the tumor remains ill-defined.
Some concerns pertain to sampling. Only female Fisher 344 rats were used in the study, and the results may not be generalizable to other species or sex. The use of a single cell line, 13762 MAT B III, to represent mammary tumors in rats may not accurately reflect the diversity of mammary tumors in the population. On another note, the use of VRPs containing coding sequences for heterologous proteins as immunotherapeutic agents may not accurately reflect the response of the immune system to naturally occurring delivery of antigens.
Technical difficulties in vaccine development are also worthy of discussion. What are the challenges in formulating a vaccine targeting a specific transcription factor and the identification of specific epitopes. Could there be potential for off-target effects and a risk for normal, healthy cells that express the germ cell-specific transcription factor, which could lead to potential side effects?
This is a worthy exploration. However, these limitations suggest that more research is necessary to fully understand the potential benefits and limitations of the therapeutic vaccine targeting BORIS for the treatment of cancer. Further studies, including human trials, are necessary to determine the safety and efficacy of the VRP-based anti-rat BORIS vaccine in humans. Until then, the conclusions should be considered preliminary and not definite.
Minor comments related to the Introduction. The Introduction could provide more background information on the field of immunotherapy and its current limitations, to provide context for the reader. Important terms such as "Immune Checkpoints Inhibitors" and "tumor-associated antigens" could be defined more clearly for the reader. The introduction could benefit from better transitions between sections, to help guide the reader through the various points being made.
Author Response
First of all, we would like to thank the reviewer for considering our manuscript, ID: ijms-2246057, "A therapeutic vaccine targeting rat BORIS (CTCFL) for treatment of rat breast cancer tumor" for publication in the International Journal of Molecular Sciences in the special issue " Tumor Microenvironment and Immune Response in Breast Cancer." As requested, we are submitting the revised manuscript, and this letter describing our responses to the concerns raised by the reviewer addressed point-by-point below.
Second reviewer:
- This study demonstrated that the VRP-BORIS vaccine was effective in inhibiting tumor growth and prolonging the survival of rats with mammary tumors. The vaccine-treated rats showed evidence of immunological memory and a stronger cellular immune response compared to control rats. However, the authors used a limited number of rats for each treatment group, which may not fully reflect the variability and heterogeneity of the disease.
This statement is correct, but if we take into account that rats were represented as a Fisher strain and the tumor is represented by a single cell line inoculated simultaneously, there is not much room for variability and heterogeneity.
- The study also only provided information about the effects on a single type of breast cancer and its effects on other types of cancer are not known.
It was beyond the scope of this study. The scope of the paper is to report proof of principle that the vaccine against single cancer-testis antigen – transcription factor Boris is sufficient to cure breast cancer in the rat model.
- Further, the study dismissed the comparison between VRP-BORIS vaccine to other treatment options (e.g. chemotherapy or radiation therapy models, or any applicable), which would provide a more comprehensive evaluation of its therapeutic efficacy in comparison to established treatments.
This animal model of BrCa has been used to test anti-BORIS vaccine efficacy. Comparison to standard treatment was beyond the scope of this study.
- Moreover, the results only show the short-term effects of the vaccine, and it is not clear if the benefits will be sustained over the long term.
Animals that were cured during vaccination were kept for three months and then checked for primary tumors and distant metastasis that appeared to be negative. We also performed experiments that are described in the paper on the re-challenge of cured rats after 144 days, and those animals not only 100% survived but also showed no signs of developing tumors, suggesting the long-term efficacy of the vaccine.
- Likewise, characterizing the specific immune response elicited by the vaccine, including which specific immune cells are activated and the types of cytokines produced, would provide additional insight into how the vaccine works and how it could be optimized. Thus, the mechanism by which the vaccine induces an immune response and eliminates the tumor remains ill-defined.
Previously we reported in a mouse model of BrCA that the anti-BORS vaccine is activating CTL that kills target cancer cells. Unfortunately, demonstrating the killing of stem cells with rats CTL is not an easy task, due in part to the very small proportion of CSCs within a tumor mass or culture.
- Some concerns pertain to sampling. Only female Fisher 344 rats were used in the study, and the results may not be generalizable to other species or sex.
The scope of the paper was to treat the rat model of breast cancer, and to our knowledge, there is no rat model for male breast cancer. Using the mouse 4T1 model, we have shown concordant results. Thus the activity of this vaccine is more likely to be 'generalizable.'
- The use of a single cell line, 13762 MAT B III, to represent mammary tumors in rats may not accurately reflect the diversity of mammary tumors in the population. On another note, the use of VRPs containing coding sequences for heterologous proteins as immunotherapeutic agents may not accurately reflect the response of the immune system to the naturally occurring delivery of antigens.
We agree with these comments. The authors were not intended to compare the natural immune response to Boris with the induced response to rat Boris upon immunization. Moreover, 100% death of the control group from inoculated tumors suggests that natural response is insufficient to cure animals, but immunization with VRP-Boris is able to cure them in up to 50% of animals. Again, we would like to stress that the scope of the paper is proof of the principle that therapeutic immunization against BORIS as monotherapy could cure a model of rat breast cancer.
- Technical difficulties in vaccine development are also worthy of discussion. What are the challenges in formulating a vaccine targeting a specific transcription factor and the identification of specific epitopes? Could there be potential for off-target effects and risk for normal, healthy cells that express the germ cell-specific transcription factor, which could lead to potential side effects? This is a worthy exploration. However, these limitations suggest that more research is necessary to fully understand the potential benefits and limitations of the therapeutic vaccine targeting BORIS for the treatment of cancer. Further studies, including human trials, are necessary to determine the safety and efficacy of the VRP-based anti-rat BORIS vaccine in humans. Until then, the conclusions should be considered preliminary and not definite.
We agree that "more research is necessary to understand the potential benefits and limitations of the therapeutic vaccine targeting BORIS for the treatment of cancer". We are planning to target BORIS cancer-testis antigen in people with stage 4 BrCA using personalized medicine.
- Minor comments related to the Introduction. The Introduction could provide more background information on the field of immunotherapy and its current limitations, to provide context for the reader. Important terms such as "Immune Checkpoints Inhibitors" and "tumor-associated antigens" could be defined more clearly for the reader. The Introduction could benefit from better transitions between sections, to help guide the reader through the various points being made.
We have significantly revised the Introduction to provide more clarity and to provide a more succinct background of the various immunotherapeutic modalities. We included brief discussions of the relative efficacy of immune checkpoint inhibitors and various tumor vaccines. We also placed this work in the context of documented mechanisms of resistance to various immunotherapies. We would propose that any reader with a basal knowledge of tumor immunology should appreciate the distinctive nature of 'tumor-associated antigens'. Nevertheless, we added a description of tumor-associated antigens. We have also revised transitions between paragraphs to ease the reader through the Introduction (see pages 1-2).
We have tried to address second reviewer's concerns fully and revised our manuscript accordingly.
If you have any questions regarding this manuscript, please contact me directly. For your convenience, my cell phone number is 301-830-0110, and my e-mail address is dloukinov@niaid.nih.gov.
Thank you for your attention to our manuscript.
Sincerely,
Dmitri Loukinov, Ph.D.
